# Comprehensive Dynamic Influence of Multiple Meteorological Factors on the Detection Rate of Bacterial Foodborne Diseases under Spatio-Temporal Heterogeneity

**DOI:** 10.3390/ijerph20054321

**Published:** 2023-02-28

**Authors:** Xiaojuan Qi, Jingxian Guo, Shenjun Yao, Ting Liu, Hao Hou, Huan Ren

**Affiliations:** 1Department of Nutrition and Food Safety, Zhejiang Provincial Center for Disease Control and Prevention, Hangzhou 310051, China; 2Zhejiang Key Laboratory of Urban Wetlands and Regional Change, Hangzhou Normal University, Hangzhou 311121, China; 3College of Resources and Environment, University of Chinese Academy of Sciences, Beijing 100049, China; 4Key Laboratory of Geographic Information Science (Ministry of Education), East China Normal University, Shanghai 200241, China; 5College of Resources, Environment and Tourism, Capital Normal University, Beijing 100048, China

**Keywords:** bacterial foodborne diseases, meteorological factors, principal component analysis, spatio-temporal scanning statistics, vector autoregressive model

## Abstract

Foodborne diseases are a critical public health problem worldwide and significantly impact human health, economic losses, and social dynamics. Understanding the dynamic relationship between the detection rate of bacterial foodborne diseases and a variety of meteorological factors is crucial for predicting outbreaks of bacterial foodborne diseases. This study analyzed the spatio-temporal patterns of vibriosis in Zhejiang Province from 2014 to 2018 at regional and weekly scales, investigating the dynamic effects of various meteorological factors. Vibriosis had a significant temporal and spatial pattern of aggregation, and a high incidence period occurred in the summer seasons from June to August. The detection rate of *Vibrio parahaemolyticus* in foodborne diseases was relatively high in the eastern coastal areas and northwestern Zhejiang Plain. Meteorological factors had lagging effects on the detection rate of *V. parahaemolyticus* (3 weeks for temperature, 8 weeks for relative humidity, 8 weeks for precipitation, and 2 weeks for sunlight hours), and the lag period varied in different spatial agglomeration regions. Therefore, disease control departments should launch vibriosis prevention and response programs that are two to eight weeks in advance of the current climate characteristics at different spatio-temporal clustering regions.

## 1. Introduction

Foodborne diseases are illnesses that result from eating food that has been contaminated with bacteria or other pathogens such as viruses or parasites [1]. Bacterial infection is the most common type of foodborne disease, especially in the summer. Foodborne diseases are a serious global health issue and cause substantial medical costs and productivity losses worldwide [2,3,4,5]. According to data collected by the Foodborne Disease Outbreaks Surveillance System during the years 2003–2017, there were 235,754 foodborne illnesses and 1457 deaths reported to the Centers for Disease Control and Prevention (CDC), China [6]. A total of 11.3% of the 13,307 outbreaks with known etiologies were caused by *Vibrio parahaemolyticus*. This ranked second among foodborne disease outbreaks after poisonous mushrooms. All types of food are potentially contaminated with bacteria. Contamination can occur during food production, harvesting, processing, storage, shipping, and preparation [7]. The source of contamination varies, and most bacteria that cause food-poisoning prefer hot and humid environments [8]. Therefore, it is meaningful to study the contributing meteorological or environmental factors of bacterial foodborne diseases for public health awareness.

Climatic factors such as temperature, humidity, and rainfall are significantly correlated with the incidence of bacterial foodborne diseases [9,10,11,12,13,14], with a certain lag effect at different time scales [15,16,17]. For example, an extreme heat wave in the high latitudes of the Arctic leads to the emergence of vibriosis [18]. The risk of *V. parahaemolyticus* infection associated with raw oyster consumption also changes with seasonal variation, time horizon, and the climate scenario [19]. The log relative risk for *Campylobacter* increased by 4.5% with the increase in weekly mean temperature in Canada [20]. There was a dynamic relationship between the outbreak of foodborne diseases and the changes in temperature and relative humidity in South Korea from 2003 to 2012, and the effects of climatic factors on eight foodborne pathogens were identified [21]. The dynamic relationship between the detection rate of bacterial foodborne diseases and climatic factors remains an important research direction in this field.

Some scholars have used time series correlation methods to analyze the incidence trends of foodborne diseases by taking into account the temporal effect in recent years [22,23,24]. Generalized autoregressive and moving average models (GSARIMA) were used to fit the number of reports of *Salmonella enterica* serovar Enteritidis cases in Sydney, Australia from 2014 to 2016 to alert future infectious outbreaks associated with high-risk foods [25]. A controlled interruption time series analysis was used to assess the impact of food safety management systems (FSMS) on foodborne disease outbreaks and food hygiene violations in Singapore [26]. The daily minimum temperature was significantly and positively correlated with salmonellosis and campylobacteriosis in South Korea on a monthly scale, with no lag period [27] found. However, these studies were usually conducted at a temporal scale of the year or month; an analysis at a finer scale may provide more information on disease control. In addition, the whole study area was modeled and analyzed equally, ignoring the characteristics of the spatial aggregation of bacterial foodborne diseases.

In contrast, spatial characteristics can help determine the epidemiological characteristics and influencing factors of the disease [28,29]. There is growing research interest in applying spatial statistical methods to identify the relationship between meteorological elements and foodborne diseases in recent years. Spatio-temporal scanning statistics were used to detect spatio-temporal clusters of bacillary dysentery in Beijing–Tianjin–Tangshan in 2011, and spatial panel models were employed to identify potential meteorological risks [30]. The geographically weighted regression method was used to determine the relationship between *V. parahaemolyticus* infection and meteorological and socioeconomic factors in the Zhejiang Province in 2018 [31]. an adaptive multigraph fusion method was used to model the effects of spatial dependencies on the risk of foodborne diseases [32]. Although these methods explored the influencing factors by considering the correlation between adjacent regions, they ignored the lagging effects of covariables such as climate factors.

A spatio-temporal model based on foodborne disease incidence data may be useful for assessing the prospective effects of climate change on food safety. Therefore, the aim of this study was to analyze the spatio-temporal pattern of bacterial foodborne disease (vibriosis) and examine the effects of meteorological factors. This study aggregated weekly cases from 2014 to 2018 in Zhejiang Province, China. A spatio-temporal scanning statistical method was used to divide the study area, and vector autoregression (VAR) models were constructed to analyze the dynamic relationship between comprehensive meteorological variables and vibriosis cases.

## 2. Methodology

### 2.1. Study Area and Data Sources

The Zhejiang Province is situated on the southeast coast of China and the southern wing of the Yangtze River Delta (Figure 1). Zhejiang is a major marine and fishery province in the country with a long history of fishing. It is located in a subtropical monsoon climate area with different types of topographies across the entire region. The high temperatures and humidity in summer and autumn are suitable for the growth and reproduction of pathogenic bacteria. The incidence of foodborne diseases in the province has increased in recent years. *V. parahaemolyticus* was the first pathogen to cause microbial foodborne disease outbreaks in Zhejiang and accounted for 58.41% of the total bacterial outbreaks [33]. The database of bacterial foodborne diseases caused by *V. parahaemolyticus* from January 2014 to December 2018 recorded the date of onset, sex, occupation, and current address. The dataset source was the Foodborne Disease Surveillance And Reporting System of Zhejiang Province, which collects case reports from 101 sentinel hospitals covering 89 county-level jurisdictions in the province. There is a high incidence of bacterial foodborne diseases in this region. However, there are few studies analyzing the dynamic relationship between bacterial foodborne diseases and meteorological factors in Zhejiang Province.

Meteorological data were provided by the China Meteorological Data Network, including county precipitation, average wind speed, average temperature, average relative humidity, daily minimum temperature, daily maximum temperature, and sunlight hours. In this study, the daily meteorological value of each station was interpolated by inverse distance weighted interpolation (IDW) and then fitted with the centroid of each county.

### 2.2. Methodological Framework

The methodological framework is illustrated in Figure 2. The spatio-temporal pattern of bacterial foodborne diseases was initially identified using a multivariate time series analysis and spatio-temporal scanning statistics. Then, the influence of high-order multicollinearity was reduced using a PCA analysis, and the effects of seven meteorological factors on bacterial foodborne diseases were explored. Finally, the VAR model was established to determine the relationship between climate variables, and the time series of bacterial foodborne diseases was examined in the different regions identified in the first step.

### 2.3. Multivariate Time Series Analysis

A time series analysis describes the changing characteristics of time series in the past and predicts the trends of the future time series; this is an important method for studying the dynamic characteristics, periodic characteristics, and correlation of statistical indicators. Scholars usually use historical data on the incidence rate of vibriosis to predict future incidence rates. However, the incidence rate is related to historical values and also depends on the values of other relevant variables in the same time series, such as temperature, relative humidity, and other climate variables. The covariance and correlation coefficients can be used to describe the degree of correlation between the two time series variables. A correlation analysis determines the interdependence between two or more variables to measure the degree and direction of the correlation between variables and the internal relationship between variables. Pearson’s correlation analysis is commonly used and is calculated as follows given a data pair of two time series variables *X* and *Y*:(1)$$P=Cor\left(X,Y\right)=\frac{Cov\left(X,Y\right)}{\sqrt{Var\left(X\right)Var\left(Y\right)}}$$

Stronger correlations are represented by larger absolute values of the correlation coefficient (correlation coefficients close to 1 or −1), while weaker correlations are represented by correlation coefficient values that are closer to 0.

### 2.4. Spatio-Temporal Scanning Statistics

Spatio-temporal scanning statistics are a common method to detect epidemic patterns of diseases at any time and space in the field of epidemiology [34]; they can be used to detect an increase in the number of local bacterial foodborne diseases and to verify whether the increase is due to random variation. The basic principle of spatio-temporal scanning statistics is to establish a spatio-temporal two-dimensional cylinder activity window based on geographical coordinates, where each cylinder reflects a possible aggregation area [35]. That is, the geographical area and time period of possible disease outbreaks. The bottom of the cylinder represents the geographical area of detection, and the height of the cylinder represents the detection time.

The expected number of cases in each scan window was calculated, and the log likelihood ratio (LLR) was used to evaluate the abnormal degree of the number of cases in the scan window based on the actual number of cases and population [36]. The relative risk (RR) was used to evaluate the risk of bacterial foodborne diseases in the cluster area [37]. The maximum likelihood ratio of all windows corresponds to the most likely cluster. The equation for determining the LLR is shown in Formula (2), and the RR of the cluster area is estimated using Formula (3) under the assumption of a Poisson distribution.
(2)$$LLR=log\left[{\left(\frac{C_{A}}{\mu_{A}}\right)}^{C_{A}}{\left(\frac{C-C_{A}}{C-\mu_{A}}\right)}^{\left(C-C_{A}\right)}\right]$$
(3)$$RR=\frac{C_{A}/\mu_{A}}{\left(C-C_{A}\right)/\left(C-\mu_{A}\right)}$$
where *C* is the total number of cases, *C_A_* is the actual number of cases in the window, and $\mu_{A}$ is the expected number of cases in the window.

### 2.5. Principal Component Analysis to Determine the Effects of Meteorological Factors

Many studies often use original meteorological data for modeling when exploring the dynamic relationship between meteorological factors and dependent variables. However, complex correlations could exist among meteorological factors, and high multicollinearity may increase the uncertainty of the analysis results. A principal component analysis is a multivariate statistical analysis method that reduces the dimension of multiple related variables to several representative comprehensive variables using linear transformations, and it contains information that is not duplicated [38]. The index is obtained after a comprehensive analysis called the main component. The weight determined by this method is based on the internal structural relationship between the indicators obtained by the data analysis; therefore, it is unaffected by subjective factors. The principal components are independent of each other, which helps reduce the interaction of information.

### 2.6. Vector Autoregressive (VAR) Model

The vector autoregressive model is suitable for multivariable time series systems to analyze the dynamic impact of random disturbances on variable systems and explain the effects of various shocks on variables [39]. The model can determine the dynamic relationship between climatic variables and the detection rate of bacterial foodborne diseases in a time series; it does not need to distinguish between exogenous and endogenous variables or add constraints to the model. The basic principle is to consider each endogenous variable in the system as a function of the lag value of all other endogenous variables in the system and construct a model. The mathematical expression of the VAR model (*p*) is
(4)$$y_{t}=A_{1}y_{t-1}+\cdots +A_{p}y_{t-p}+B_{x_{t}}+\varepsilon_{t}\;\left(t=1,2,\cdots ,T\right)$$
where *y_t_* is the k-dimensional endogenous variable vector, *x_t_* is the d-dimensional exogenous variable vector, $\varepsilon_{t}$ is the random disturbance vector, *p* is the lag order, *T* is the number of bacterial foodborne disease case samples, and *A*_1_*…A_p_*, and *B* represents the *k × k* dimensional coefficient matrix that requires estimation.

The construction of the VAR model mainly includes the following steps:

Unit root test of sample data: It is necessary to test the unit root of the time series before establishing the model to ensure the stationarity of the time series and avoid the pseudo-regression phenomenon. The augmented Dickey–Fuller test (ADF test) was used to evaluate the unit root [40], and the null hypothesis was that there was a unit root. There is no unit root if the sequence is stable following this test. The *t*-statistic is under three confidence levels (1%, 5%, and 10%), and the original hypothesis is rejected; otherwise, there is a unit root.

The optimal lag order was determined and a VAR model was constructed: A greater order represents a greater degree of freedom for the model. However, a larger order indicated that more parameters required estimation. This partly affected the validity of the model. Therefore, there should be enough lag terms and a sufficient number of degrees of freedom when determining the optimal lag order. This is mainly based on Akaike information criterion (AIC) and Schwarz Criterion (SC), which state that a model with smaller statistics has a better fitting effect. The stationarity test and fitting effect of the model are good if the eigenvalues of each variable calculated by the model fall in the unit circle.

Granger causality test: The causal relationship between variables was examined by testing whether the lagging term of one variable has an impact on the other variables.

Impulse response: The impulse response function is mainly used to analyze the dynamic effects of the random disturbance of each endogenous variable on itself and all other endogenous variables.

## 3. Results

### 3.1. Multivariate Time Series Correlation Analysis

A total of 182,473 samples of bacterial foodborne diseases were collected from 2014 to 2018 in the Zhejiang Province. There were 6430 (3.52%) positive cases due to *V. parahaemolyticus* infection, and 6226 of 6430 positive cases occurred between the months of May and October during these 4 years. Table 1 summarizes the descriptive statistics for the weekly detection rate of *V. parahaemolyticus* and the ambient meteorological conditions on the detection date. Time sequence diagrams were used to visually show temporal trends in pathogen infection and meteorological variations (Figure 3). The *V. parahaemolyticus* infection showed an annual peak during the summer that was consistent with the temperature changes. Moreover, the correlation analysis showed that *V. parahaemolyticus* was closely correlated with all seven climatic factors, but showed a time lag (Figure 4). Six climatic factors had a positive lag effect on pathogen detection, with different lag periods. The duration of sunlight was over two lag weeks, the temperature was over three lag weeks, and the relative humidity and precipitation were over eight weeks. Meanwhile, wind speed (lag of 11 weeks) had a weak negative correlation (−0.171) with vibriosis.

The *PCA* model was established based on the seven meteorological indicators *X*_1_, *X*_2_, …, *X*_7_ to reduce the interference of multicollinearity. The cumulative variance contribution rate of the three principal components (*F*_1_, *F*_2_, and *F*_3_) was 90.377%, which meets the requirement of a cumulative variance of 0.85. A further quantitative analysis of the climatic factors showed that the first principal component reflecting temperature-related climate parameters (including mean temperature, daily maximum temperature, and daily minimum temperature) had a variance contribution rate of 46.454%. The second principal component reflects the climate parameters related to water (including average relative humidity, precipitation, and sunlight hours) and had a variance contribution rate of 29.505%. The third principal component was the wind-related climate parameters (including the average wind speed), with a variance contribution rate of 14.418%. The relationships between the principal components and original climate variables are shown in Table 2.

### 3.2. Spatio-Temporal Scanning Statistics of the Detection Rate of the V. parahaemolyticus

Three significant spatio-temporal aggregation areas were detected using spatio-temporal scanning statistics (Table 3). Vibriosis had an evident clustering tendency in time and space. The period from the 28th to the 37th week of 2016 (June–August 2016) had the highest incidence of vibriosis. The detection rate of *V. parahaemolyticus* in the eastern coastal areas and northwest Zhejiang Province was significantly higher than that in the southwest area from a spatial perspective (Figure 5).

### 3.3. VAR Model Fitting and Prediction

The study area was divided into four regions based on the three clustered areas and one non-clustered area. Four VAR models were constructed to measure the internal relationship between the detection rate of *V. parahaemolyticus* and the principal components of the three meteorological factors. It is necessary to conduct unit root and cointegration tests before establishing the model to ensure the stationarity of the time series and avoid pseudo-regression. The unit root tests showed that the original sequences of the detection rate of *V. parahaemolyticus* and the components of meteorological factors in the four regions were a stationary series (Table 4). This was in accordance with the premise of establishing the VAR model.

The three meteorological principal components of the four regions were introduced into the four VAR models as explanatory variables. The information of the AIC and SC were combined to determine that the optimal lag periods of *C*_1_, *C*_2_, *C*_3_, and *Non_C* were 6, 2, 6, and 2 weeks, respectively.

There was a significant one-way Granger causal relationship between the detection rate (*C*_1_*_DR*) and the first meteorological principal component (*C*_1_*_F*_1_) or meteorological principal component 3 (*C*_1_*_F*_3_) for *C*_1_ (Table 5). This indicated that changes in climate-related parameters of temperature and wind speed significantly changed the detection rate of *V. parahaemolyticus*. Only the first meteorological principal component (*C*_2_*_F*_1_) significantly increased the detection rate for *C*_2_, *C*_3_, or *Non_C*. This indicated that changes in climate-related parameters of temperature could lead to corresponding changes in the detection rate of *V. parahaemolyticus* in these areas.

All four models treated the local detection rate of *V. parahaemolyticus* as the dependent variable and the principal components of the three meteorological factors as the explanatory variables to form the regression function. The formulae were as follows:(5)$$C_{1}\_DR\left(6\right)=0.0008C_{1}\_F_{1}\left(-1\right)+0.0017C_{1}\_F_{1}\left(-2\right)+0.0008C_{1}\_F_{1}\left(-3\right)+0.0022C_{1}\_F_{1}\left(-4\right)+0.004C_{1}\_F_{1}\left(-5\right)\;+0.0023C_{1}\_F_{1}\left(-6\right)-0.0024C_{1}\_F_{2}\left(-1\right)-0.0017C_{1}\_F_{2}\left(-2\right)+0.0016C_{1}\_F_{2}\left(-3\right)-0.0017C_{1}\_F_{2}\left(-4\right)\;-0.0018C_{1}\_F_{2}\left(-5\right)-6.3018e^{-6}C_{1}\_F_{2}\left(-6\right)+0.0068C_{1}\_F_{3}\left(-1\right)-0.0005C_{1}\_F_{3}\left(-2\right)-0.0009C_{1}\_F_{3}\left(-3\right)\;+0.0063C_{1}\_F_{3}\left(-4\right)+0.0018C_{1}\_F_{3}\left(-5\right)-0.0057C_{1}\_F_{3}\left(-6\right)+0.33C_{1}\_DR\left(-1\right)+0.2732C_{1}\_DR\left(-2\right)\;+0.0606C_{1}\_DR\left(-3\right)-0.0159C_{1}\_DR\left(-4\right)+0.0020C_{1}\_DR\left(-5\right)-0.1673C_{1}\_DR\left(-6\right)+0.0155$$
(6)$$C_{2}\_DR\left(2\right)=0.0141C_{2}\_F_{1}\left(-1\right)+0.0073C_{2}\_F_{1}\left(-2\right)+0.0014C_{2}\_F_{2}\left(-1\right)-0.0026C_{2}\_F_{2}\left(-2\right)+0.0078C_{2}\_F_{3}\left(-1\right)\;+0.0016C_{2}\_F_{3}\left(-2\right)+0.2527C_{2}\_DR\left(-1\right)+0.0787C_{2}\_DR\left(-2\right)+0.0299$$
(7)$$C_{3}\_DR\left(6\right)=0.0084C_{3}\_F_{1}\left(-1\right)+0.0023C_{3}\_F_{1}\left(-2\right)-0.0059C_{3}\_F_{1}\left(-3\right)+0.0003C_{3}\_F_{1}\left(-4\right)+0.0005C_{3}\_F_{1}\left(-5\right)\;+0.0012C_{3}\_F_{1}\left(-6\right)+0.0006C_{3}\_F_{2}\left(-1\right)-0.0006C_{3}\_F_{2}\left(-2\right)+6.2861e^{-5}C_{3}\_F_{2}\left(-3\right)-0.0028C_{3}\_F_{2}\left(-4\right)\;+0.0002C_{3}\_F_{2}\left(-5\right)-0.0003C_{3}\_F_{2}\left(-6\right)+0.0013C_{3}\_F_{3}\left(-1\right)-0.0015C_{3}\_F_{3}\left(-2\right)-0.002C_{3}\_F_{3}\left(-3\right)\;-0.0019C_{3}\_F_{3}\left(-4\right)-0.0039C_{3}\_F_{3}\left(-5\right)+0.0018C_{3}\_F_{3}\left(-6\right)+0.3676C_{3}\_DR\left(-1\right)+0.2425C_{3}\_DR\left(-2\right)\;+0.0215C_{3}\_DR\left(-3\right)-0.0386C_{3}\_DR\left(-4\right)+0.1947C_{3}\_DR\left(-5\right)-0.1414C_{3}\_DR\left(-6\right)+0.008\;\;$$
(8)$$\mathrm{Non}\_C\_DR\left(2\right)=0.0027\mathrm{Non}\_C\_F_{1}\left(-1\right)+0.0019\mathrm{Non}\_C\_F_{1}\left(-2\right)+0.0011\mathrm{Non}\_C\_F_{3}\left(-1\right)-0.0007\mathrm{Non}\_C\_F_{3}\left(-2\right)\;+0.3699\mathrm{Non}\_C\_DR\left(-1\right)+0.1301\mathrm{Non}\_C\_DR\left(-2\right)-0.0013\mathrm{Non}\_C\_F_{2}+0.0074$$

In general, all four VAR models performed well, with adjusted *R^*2*^* values ranging from 0.573 to 0.727 (Table 6). All unit roots of the four models were in the unit circle. This indicated that the structure of the model was stable.

### 3.4. Impulse Response Analysis of V. parahaemolyticus Detection

The impulse response analysis provided insight into the dynamic reactions between meteorological parameters and the detection rate of *V. parahaemolyticus* [41]. Figure 6 illustrated that the impulse response function amounts to one standard deviation of the endogenous variable. In the first column, the response of detection rate to the temperature-related principal component (F_1_) shock showed similar performance in the *C*_1_, *C*_2_, and *C*_3_ clusters, while the magnitude was slightly smaller in the Non_C region. The responses to temperature-related component shocks were all positive, indicating that temperature significantly increased the detection rate. This promoting effect showed a significant increase in the first 3 weeks, then slowly increased and stabilized, and finally gradually weakened. The positive effect of ambient temperature on the detection rate of *V. parahaemolyticus* was more significant in the range of *C*_1_; the continuous growth time was longer (up to 7 weeks). The detection rate of *V. parahaemolyticus* slightly decreased in *C*_3_ from 3 to 4 weeks, and the overall trend was like that observed in *C*_2_. In other words, it gradually stabilized after 3 weeks.

The impulse responses to the moisture-related principal component (F_2_) shock slightly fluctuated around 0 in *C*_1_, *C*_2_, and *C*_3_ in the first 7 weeks, which was consistent with the results of the Granger causality test with lag 2 and 6 (Table 5); they exhibited a slightly positive effect after 7 weeks. The moisture variable was exogenous in the model of the *Non_C* region; therefore, it did not participate in the impulse response test.

Similarly, the magnitude of the responses to the wind-related principal component (F_3_) shock was smaller than F_1_. To a shock of the wind-related principal component, the response of the detection rate of *C*_1_ positively fluctuated until the 6th week; however, the persistence was the smallest among all meteorological principal components.

## 4. Discussion

The risk of vibriosis in Zhejiang Province from 2014 to 2018 was significantly aggregated in the spatial and temporal dimensions. The incidence was high from June to August, and the detection rate of *V. parahaemolyticus* in the eastern coastal areas and northwest was significantly higher than in the southwest. This may be related to the high-temperature and humid environment in the Zhejiang Province during this period. This climate condition is conducive to the growth and reproduction of microorganisms such as *V. parahaemolyticus.* Thus, food is prone to spoilage. Differences in regional eating habits and economic conditions may be the key factors leading to spatial differences in bacterial foodborne diseases. The three high agglomeration areas of *C*_1_, *C*_2_, and *C*_3_ belong to areas with relatively high per capita disposable income (typical representatives of *Ningbo*, *Hangzhou*, and *Wenzhou*). The residents’ ability to consume food outside the home is relatively high. Aquatic products and seafood were the first food species that were suspected to cause *V. parahaemolyticus* infection [42]. The eastern region (*C*_1_ and *C*_2_) is close to the port and wharf and is rich in aquatic products. Local residents live near the sea and enjoy eating seafood and aquatic products. This greatly increases the infection rate of *V. parahaemolyticus*. Therefore, food safety inspection departments in coastal areas need to standardize and strengthen the routine hygiene testing of aquatic products and seafood in the market, especially during the high-incidence summer period. The market supervision department needs to strengthen the hygiene supervision of various restaurants and the training of employees in standard operations. In addition, residents in coastal areas require public awareness and education on food hygiene and the prevention of bacterial foodborne diseases.

The detection rate of *V. parahaemolyticus* in Zhejiang Province may have resulted from the comprehensive effect of climate components based on the PCA and VAR models in this study. Specifically, the occurrence of this bacterial foodborne disease is directly and critically affected by temperature. The influence of moisture variables such as precipitation, relative humidity, and sunlight hours was more complex, and the effect on the model was not prominent. In addition, the comprehensive effect of climate variables had a lagging effect on the infection of bacterial foodborne diseases, and the lagging effect varied between regions. These two findings are consistent with those of Lake et al. in England and Wales [43] and Zhang et al. in Australia [14]. The impulse response function analysis of the four VAR models showed that the temperature meteorological parameters had a positive effect on vibriosis in all four areas of Zhejiang Province. The impact gradually increased and stabilized over time. Previous studies showed a positive correlation between temperature and the occurrence of bacterial foodborne diseases [44,45,46]. In other words, a higher-temperature environment promotes the reproduction of *V. parahaemolyticus* and increases the probability of food infection, resulting in a positive impact on the infection rate of the disease. Therefore, the detection rate of *V. parahaemolyticus* significantly increased in the early stage of temperature change. The impulse response results showed that the linkage relationship between the temperature and detection rate varied in different regions. In general, it is difficult for people to promptly respond to sudden changes in temperature, which increases the risk of infection. However, it will raise their awareness of bacterial foodborne diseases and food deterioration and may reduce the occurrence of disease infection once people adapt to the environment, such as in the case of long-term heat waves. The modeling showed that there were regional differences in the response characteristics of bacterial foodborne diseases to temperature variables. The degree of responses of the infection probability to temperature variables in the aggregation areas were significantly higher than that of the non-cluster area, with a fast growth rate and long duration. The impact of temperature on *V. parahaemolyticus* infection in *C*_1_ was characterized by a long duration (approximately 7 weeks) and strong response intensity (the maximum was approximately 0.011). *C*_2_ was characterized by a large response intensity (the maximum was approximately 0.0125), whereas *C*_3_ was relatively weak (the maximum was approximately 0.008). The response intensity of *Non_C* to temperature infection in the non-clustered area of bacterial foodborne diseases was significantly lower than the average value (the maximum value was approximately 0.004). These results showed that the sub-regional model avoided the mutual interference of various factors between regions and maximized the comprehensive effect of meteorological factors on the detection rate of *V. parahaemolyticus*.

The positive response to wind-related principal component shock showed that wind speed can affect the reproduction of *V. parahaemolyticus*, viability, and time in the environment. Weather conditions with high wind speeds may be conducive to the spread of bacterial pollutants and may indirectly affect the occurrence of *Vibriosis*; this is consistent with previous findings [47]. The impulse response of detection rate of *V. parahaemolyticus* to the moisture-related principal components was not obvious in the initial stage. The growth trend of the impulse response curve showed that the moisture meteorological parameters had a lagging effect on vibriosis after being influenced for a period of 7 weeks. Precipitation in the time series diagram showed an obvious lag effect, and the lag response in different regions was different (Figure 7). For example, the detection peak of *V. parahaemolyticus* in Zhejiang Province corresponded to the detection peak of *V. parahaemolyticus* after 3–7 weeks. This was consistent with the impulse response of the moisture meteorological parameters in this model. However, it differs from previous studies. For example, there is a significant positive correlation between vibriosis and relative humidity in Korea [21]. Meanwhile, changes in the meteorological conditions (such as humidity) may change the characteristics of survival and transmission patterns of microorganisms such as *V. parahaemolyticus*, increasing the number of cases of vibriosis [48]. This difference may be due to those studies failing to consider the combined effects of meteorological factors and spatial heterogeneity. These results indicated spatial differences in the effects of meteorological factors on the detection rate of bacterial foodborne diseases. This reflected the effectiveness of the innovative idea of dividing the study area with spatio-temporal scanning statistics to explore the dynamic relationship between detection rates and meteorological factors.

## 5. Conclusions

This study proposed a framework to evaluate the dynamic relationship between different meteorological factors and the risk of bacterial foodborne diseases based on the spatio-temporal heterogeneity of vibriosis in Zhejiang Province. A VAR model was combined with foodborne diseases to analyze the spatio-temporal risk characteristics and potential climate risk factors of bacterial foodborne diseases.

Vibriosis exhibited significant temporal and spatial aggregation and peaked in the summer. The detection rate of *V. parahaemolyticus* was significantly higher in the eastern coastal areas and northwest compared with the southwest. In addition, temperature, relative humidity, precipitation, sunlight hours, and wind speed were important meteorological indexes affecting vibriosis that lag by 3, 8, 8, 2, and 11 weeks, respectively. The comprehensive dynamic influence of the meteorological factors showed regional variation. The temperature-related principal component had a stronger promoting effect on the detection rate in spatio-temporal cluster areas, while the influence in the non-clustered area was relatively weak. The zoning modeling framework proposed in this study considered the temporal aggregation effect of vibriosis and the spatial agglomeration characteristics of epidemics from the spatial dimension compared with whole region modeling as conducted in previous studies; this maximized the impact of reduced climate variables on regional foodborne disease infection probability. This study provided guidance and geographical support for relevant government departments to prevent and control foodborne diseases in Zhejiang Province. This research framework can be extended to other problems caused by climate and environmental change.

The following limitations and future perspectives were proposed. First, the modeling area in this study was based on the administrative region and was divided according to the results of spatio-temporal scanning statistics. This method ignored differences in social and economic conditions and may be inadequate. For example, the 2017 per capita GDP in Haining City and Taishun County was CNY 103,220 and CNY 36,420, respectively. The gross domestic product of the two places was quite different, but it was divided into non-clustered areas (*Non_C*) in the spatio-temporal scanning results. An understanding of regional divisions, conducted in a more reasonable manner, will play an important role in future studies. Second, the evaluation framework for the drivers of foodborne disease incidence must be further improved. This study only considered meteorological factors in the modeling; thus, future studies should include indicators such as food consumption structure, food exposure information, age structure, and financial and health expenditure to explore the factors influencing the incidence of bacterial foodborne diseases under spatio-temporal heterogeneity from a more comprehensive point of view.

## Figures and Tables

**Figure 1 ijerph-20-04321-f001:**
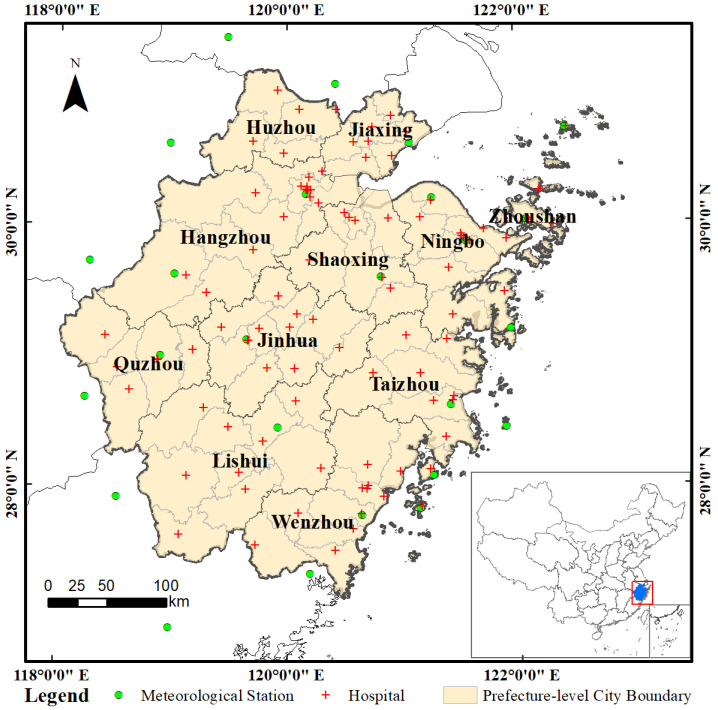
Geographical location of the study area showing the distribution of meteorological stations and hospitals.

**Figure 2 ijerph-20-04321-f002:**
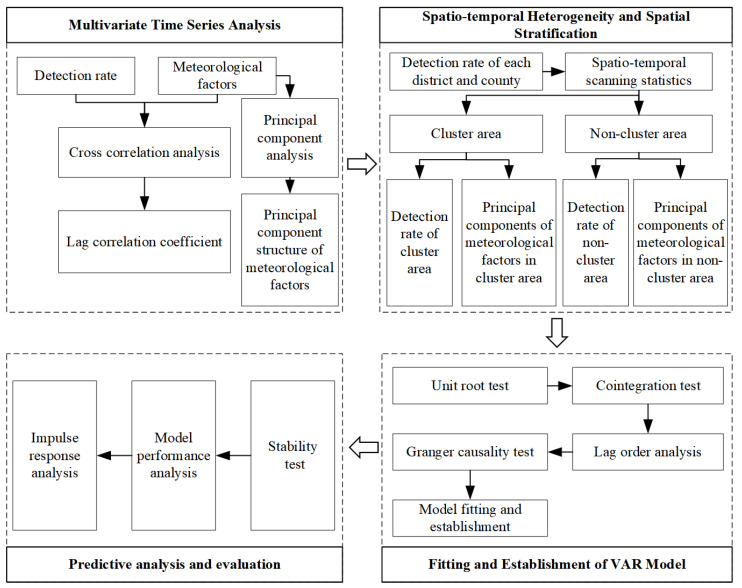
Methodological framework.

**Figure 3 ijerph-20-04321-f003:**
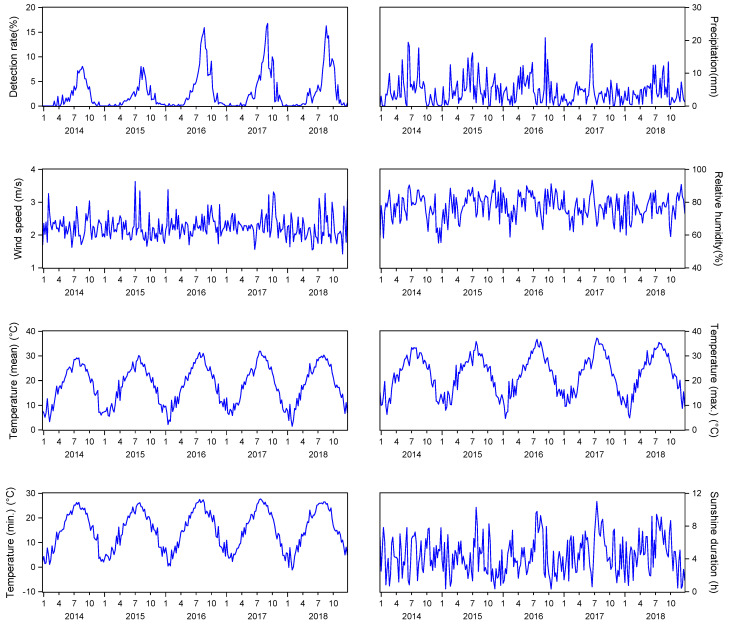
Time trend for *V. parahaemolyticus* infection and average daily precipitation, wind speed, relative humidity, temperature (mean, maximum, and minimum), and sunlight duration in the Zhejiang Province, 2014–2018.

**Figure 4 ijerph-20-04321-f004:**
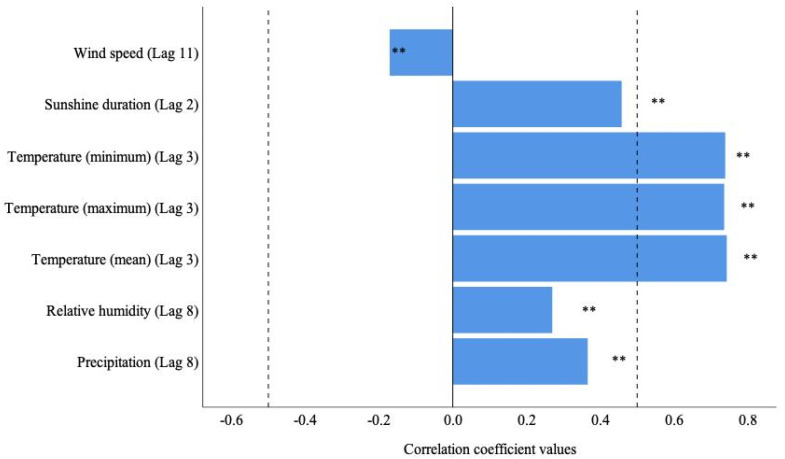
Correlations between seven climatic factors and the weekly detection rate of *V. parahaemolyticus* in Zhejiang Province from 2014 to 2018. Lag number: number of weeks prior; ** *p* < 0.01.

**Figure 5 ijerph-20-04321-f005:**
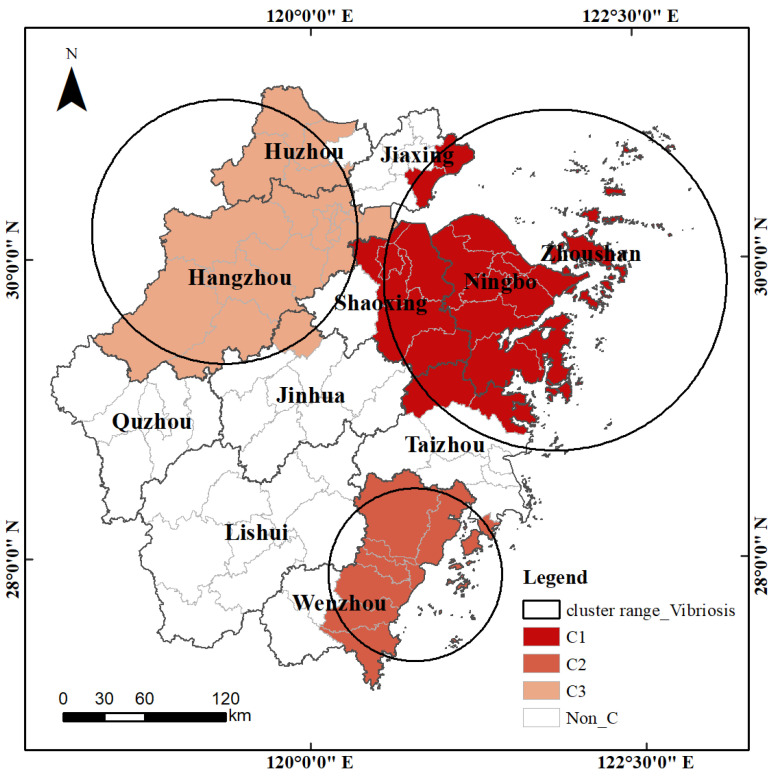
Spatio-temporal clustering characteristics of the detection rate of *V. parahaemolyticus*.

**Figure 6 ijerph-20-04321-f006:**
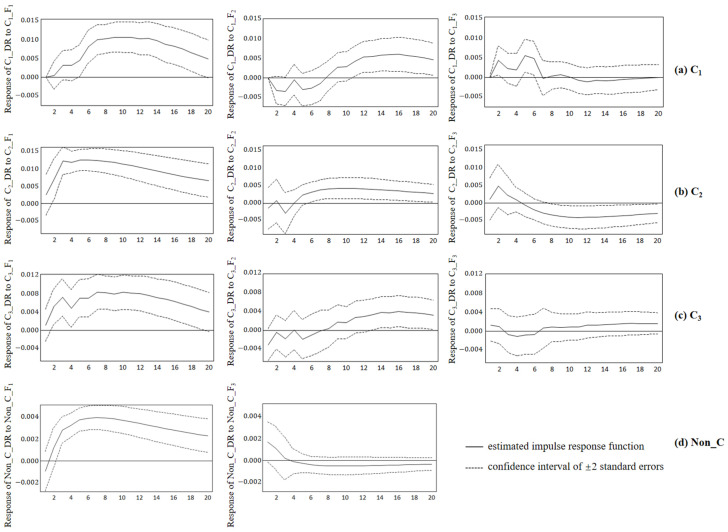
Responses of the detection rate to a meteorological component shock in (**a**) the C_1_ cluster, (**b**) C_2_ cluster, (**c**) C_3_ cluster, and (**d**) Non_C region at given horizon h (h = 1, 2, … 20) with a confidence band. Notes: The forecast horizon [weeks] is given on the horizontal axis. The vertical axis shows the magnitude of the impulse response, and a negative value represents the reduction effect. Dashed lines represent a confidence interval of ±2 standard errors.

**Figure 7 ijerph-20-04321-f007:**
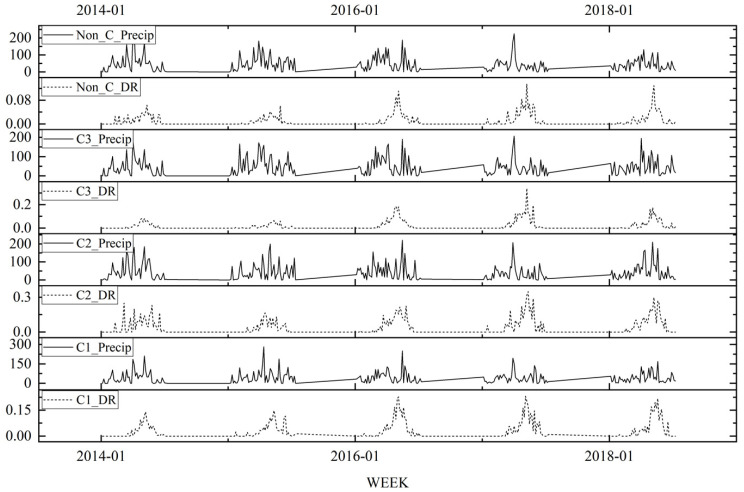
Time series of the weekly precipitation and detection rate of *V. parahaemolyticus*.

**Table 1 ijerph-20-04321-t001:** Summary of the weekly detection rate of *V. parahaemolyticus* and weather conditions of Zhejiang province, 2014–2018.

Variables	Description	n	Minimum	Maximum	Median	Mean	S.D.
Dependent variable							
DR	Detection rate of *V. parahaemolyticus* (%)	261	0.00	16.79	0.99	2.70	3.70
Meteorological characteristic							
SunHour	Sunshine hours (h)	261	0.34	11	4.22	4.44	2.33
MaxTemp	Daily maximum temperature (°C)	261	4.63	37.24	23.62	22.48	8.08
MinTemp	Daily minimum temperature (°C)	261	−1.13	27.71	15.22	15.05	7.94
MeanTemp	Mean temperature (°C)	261	1.44	31.91	18.68	18.2	7.94
MeanHum	Mean relative humidity (%)	261	55.16	93.4	77.59	77.16	7.64
MeanWS	Mean wind speed (m/s)	261	1.42	3.63	2.19	2.24	0.36
Precip	Precipitation (mm)	261	0.00	20.77	3.61	4.48	4.08

**Table 2 ijerph-20-04321-t002:** Component matrix and interpretation of total variance.

Climate Index	Variables	*F* _1_	*F* _2_	*F* _3_
Precip	*X* _1_	0.346	0.751	0.034
MeanWS	*X* _2_	−0.042	−0.049	0.997
MeanHum	*X* _3_	0.283	0.858	0.076
MeanTemp	*X* _4_	0.993	−0.066	0.003
MaxTemp	*X* _5_	0.980	−0.163	−0.055
MinTemp	*X* _6_	0.988	0.026	0.040
SunHour	*X* _7_	0.356	−0.855	0.061
Eigenvalue		3.252	2.065	1.009
% of variance		46.454	29.505	14.418
% of cumulative variance		46.454	75.959	90.377

**Table 3 ijerph-20-04321-t003:** Spatio-temporal aggregation characteristics of vibriosis in Zhejiang Province from 2014 to 2018.

Cluster	Duration (Weeks)	Number of Countries	*RR*	*LLR*
C_1_ ***	29th–40th, 2016	23	4.610	390.316
C_2_ ***	22nd, 2016–42nd, 2017	10	3.470	372.045
C_3_ ***	28th–37th, 2016	18	3.700	190.408
Non_C	-	38	-	-

Non_C, non-clustered area; *** *p* < 0.001; RR, relative risk; LLR, log likelihood ratio.

**Table 4 ijerph-20-04321-t004:** Unit root test results.

Types	Variables	Difference Order	Exogenous	ADF Test *t*-Statistic	Test Critical Values			Conclusion
					1%	5%	10%	
Detection rate	C_1__DR	0	None	−2.922	−2.574	−1.942	−1.616	Steady
	C_2__DR	0	Constant	−3.882	−3.456	−2.873	−2.573	Steady
	C_3__DR	0	Constant	−4.048	−3.455	−2.872	−2.573	Steady
	Non_C_DR	0	Constant	−4.934	−3.455	−2.872	−2.573	Steady
Meteorological component	C_1__F_1_	0	None	−7.378	−2.574	−1.942	−1.616	Steady
	C_1__F_2_	0	None	−11.787	−2.574	−1.942	−1.616	Steady
	C_1__F_3_	0	Constant	−14.374	−3.455	−2.872	−2.573	Steady
	C_2__F_1_	0	None	−7.894	−2.574	−1.942	−1.616	Steady
	C_2__F_2_	0	None	−11.980	−2.574	−1.942	−1.616	Steady
	C_2__F_3_	0	Constant	−7.166	−3.455	−2.872	−2.573	Steady
	C_3__F_1_	0	None	−5.296	−2.574	−1.942	−1.616	Steady
	C_3__F_2_	0	None	−11.859	−2.574	−1.942	−1.616	Steady
	C_3__F_3_	0	Constant	−14.696	−3.994	−3.427	−3.137	Steady
	Non_C_F_1_	0	None	−7.831	−2.574	−1.942	−1.616	Steady
	Non_C_F_2_	0	None	−11.811	−2.574	−1.942	−1.616	Steady
	Non_C_F_3_	0	Constant	−14.661	−3.455	−2.872	−2.573	Steady

**Table 5 ijerph-20-04321-t005:** The result of variables in the Granger causality test.

Region	Null Hypothesis	F-Statistic	*p*-Values	Conclusion
C_1_	C_1__F_1_ does not Granger Cause C_1__DR ***	6.66244	2 × 10^−6^	Reject
	C_1__F_2_ does not Granger Cause C_1__DR	0.73259	0.6238	Accept
	C_1__F_3_ does not Granger Cause C_1__DR *	2.18704	0.0449	Reject
C_2_	C_2__F_1_ does not Granger Cause C_2__DR ***	29.2360	4 × 10^−12^	Reject
	C_2__F_2_ does not Granger Cause C_2__DR	0.96566	0.3821	Accept
	C_2__F_3_ does not Granger Cause C_2__DR	0.44662	0.6403	Accept
C_3_	C_3__F_1_ does not Granger Cause C_3__DR ***	5.10501	6 × 10^−5^	Reject
	C_3__F_2_ does not Granger Cause C_3__DR	1.57484	0.1551	Accept
	C_3__F_3_ does not Granger Cause C_3__DR	0.30514	0.9339	Accept
Non_C	Non_C_F_1_ does not Granger Cause Non_C_DR***	19.4235	1 × 10^−8^	Reject
	Non_C_F_2_ does not Granger Cause Non_C_DR	0.19730	0.8211	Accept
	Non_C_F_3_ does not Granger Cause Non_C_DR	0.42030	0.6573	Accept

** p* < 0.05; *** *p* < 0.001.

**Table 6 ijerph-20-04321-t006:** Model performance of the VAR analysis for four regions in the Zhejiang Province from 2014 to 2018.

Evaluation Index	C_1__DR (6)	C_2__DR (2)	C_3__DR (6)	Non_C_DR (2)
R-squared	0.753	0.584	0.667	0.585
Adj. R-squared	0.727	0.571	0.632	0.573
AIC	−4.278	−3.229	−4.253	−5.580
SC	−3.931	−3.105	−3.905	−5.470

AIC, Akaike information criterion; SC, Schwarz criterion.

## Data Availability

The data used to support the findings of this study are available from the corresponding author upon request.

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
