# Peer review of "Comprehensive Dynamic Influence of Multiple Meteorological Factors on the Detection Rate of Bacterial Foodborne Diseases under Spatio-Temporal Heterogeneity"

_ijerph, 2023, doi:10.3390/ijerph20054321_

Round 1

Reviewer 1 Report

The article is well written, makes a relevant contribution to the topic, using statistical analyzes that are complex for the general public to understand. However, from the point of view of the final modeling (VAR model construct) there is a problem regarding the statistical distribution imposed on the response variable (DR - Detection rate of V. parahaemolyticus), because as can be seen in Figure 5 Impulse response function analysis of VAR the results of the simulations carried out also imply negative estimated DR, which does not make sense, because in this case, as it is a rate, the result is always greater than or equal to zero. Therefore, we request that the construction of the Vector autoregressive (VAR) model use a probability distribution that considers the specificity of the response variable.

In Figure 5, it also seems strange to apply a shock to the principal components F1, F2, and F3, because, although their use facilitates modeling, they have no practical application, since it is not known what an increase of 1 unit means on F1, or on F2, or on F3.

Another question is why keep principal components that were not significant in the final analysis? In our view this makes no sense. If it was not significant, it is better to disregard it in the final analysis, in order to facilitate the interpretation and also to reduce the computational work to obtain the estimates.

Other required corrections are given below:

Lines 55, 64, 67, 71, 77, 82, 433, 434: Delete the comma (“,”) before “et al”.

Line 250, Figure 4: Fix the word “Corelation” in x-axis. The correct is “Correlation”.

Line 355-356, Figure 5. It is not clear what the curves represent (blue line and orange dots?). Why is the Non_C_DR and Non_C_F2 response graph not displayed? In the Response C1_DR to C1_F1 graph, part of the orange dots are not displayed.

Author Response

Thank you for your careful review. We really appreciate your efforts in reviewing our manuscript. Please find the attached file for the response to comments.

Reviewer 2 Report

Title: Comprehensive dynamic influence of multiple meteorological factors on the detection rate of bacterial foodborne diseases under spatio-temporal heterogeneity

Manuscript Number        : ijerph-2163920

The manuscript entitled “Comprehensive dynamic influence of multiple meteorological factors on the detection rate of bacterial foodborne diseases under spatio-temporal heterogeneity is a more efficient research work on bacterial foodborne diseases.

The authors have surveyed and estimated the Spatio-temporal heterogeneity of multiple meterological factors on the detection rate of vibrio parahaemolyticus in eastern coastal areas and northwestern Zhejiang. The author given an important message to disease control departments to conduct disease control and response programs for two to eight weeks in advance, for climatic conditions at Spatio-temporal heterogeneity regions. I highly recommend the journal editor for publication. The manuscript needs to improve its English, and the information presented in this paper is detailed and precise. I recommend the authors undergo a thorough minor review of the manuscript for alignment corrections.

Abstract:

Author has to modify the font size “of bacterial foodborne diseases”

Lane 34: replace the sentence “food contaminated” in to “food that has been contaminated”

Conclusion:

The author must provide a strong conclusion point and address other issues 

Author Response

Thank you for your careful review. We really appreciate your efforts in reviewing our manuscript. Please find the attached file for the response to your kind comments.

Reviewer 3 Report

In this study, Qi et al. report considering the spatio-temporal heterogeneity, we analyzed the spatiotemporal patterns of vibriosis in Zhejiang Province from 2014 to 2018 at regional and weekly scales and investigated the dynamic effects of various meteorological factors.. These findings look interesting and novel. However, some issues need to be addressed before further consideration.

1.      When the strain name appears for the first time, it should be written in full Latin, and when it appears for the second time, it should be written in short Latin.

2.      Introduction: How about the novelty of the current study comparing with previous studies? Relevant information should be emphasized in the paper.

3.      There are numerous grammatical mistakes in the manuscript, Please have the text edited by a native English speaker.

4.      The reference format of this manuscript needs to be unified.

Author Response

(The authors gave the same response as above.)

Reviewer 4 Report

This study evaluated the temporal and spatial associations between meteorological factors and vibriosis risks. I am impressed with the datasets and the well-developed model. Overall, the manuscript is logical and well written. I would recommend acceptance if some minor issues are fixed.

1. The font of some text is not consistent. For example: line19: “of bacterial foodborne diseases” is larger than other words. The same can be found in lines 48-49, 51, 60-64, 73, 89, 91, 98, 101-103, 106-107, 135-137, 142-144, 160, 166, 176, 180-183, 199-200, 206-209, 212-216, etc., in the introduction and methods parts. I will not list any in the results, discussion, and conclusion sections.

2. There are also some grammatic errors. The tense is not consistent. I suggest the authors to carefully read and double check the grammar.

3. I suggest the authors to move the titles of each subfigure to the right close to the y-axis. It would make the figures more reader friendly.

4. Figure 5 is not clear. The authors should clarify what x-axis, blue line, and orange lines are as well as move the subtitle besides the y-axis.

Author Response

(The authors gave the same response as above.)
